# Significant Impact of Growth Medium on Itraconazole Susceptibility in Azole-Resistant Versus Wild-Type *Trichophyton indotineae, rubrum,* and *quinckeanum* Isolates

**DOI:** 10.3390/ijms26157090

**Published:** 2025-07-23

**Authors:** Luisa Krauße, Anke Burmester, Silke Uhrlaß, Mario Fabri, Pietro Nenoff, Jörg Tittelbach, Cornelia Wiegand

**Affiliations:** 1Department of Dermatology, Jena University Hospital, Friedrich Schiller University, D-07747 Jena, Germanymario.fabri@med.uni-jena.de (M.F.); joerg.tittelbach@med.uni-jena.de (J.T.); c.wiegand@med.uni-jena.de (C.W.); 2Labopart-Medizinische Laboratorien, D-04571 Rötha, Germanyp.nenoff@labopart.de (P.N.)

**Keywords:** ergosterol biosynthesis, squalene epoxidase, *Erg1*, sterol 14-α demethylase *Erg11* itraconazole, gene amplification

## Abstract

Azole resistance in dermatophytes, particularly *Trichophyton indotineae*, has become a growing global concern. Current antifungal susceptibility testing protocols (EUCAST, CLSI) have limitations in reproducibility and sensitivity. This study aimed to evaluate how medium composition, incubation temperature, and spore concentration influence itraconazole susceptibility testing across various dermatophyte species. Thirty-eight clinical isolates representing *Trichophyton*, *Microsporum*, and *Epidermophyton* species were tested using a microplate laser nephelometry system (MLN). IC_50_ values for itraconazole were determined in three different media (Sabouraud glucose (SG), RPMI-based (RG), and RG supplemented with casein (RGC)) at 28 °C and 34 °C. Effects of spore concentration on growth dynamics and lag phase were also analyzed. SG medium provided clear phenotypic separation between resistant and sensitive isolates. In contrast, RG and RGC showed overlapping IC_50_ values. Lower spore concentrations revealed underlying growth differences, which were masked at higher inoculum levels. Temperature and media composition significantly affected IC_50_ outcomes. Genotypic analysis confirmed resistance-associated *Erg11B* point mutations and genomic amplifications in *T. indotineae*, particularly in combination with *Erg1* mutations, forming distinct subpopulations. SG medium combined with reduced spore concentrations offered improved differentiation of resistant versus sensitive strains. These findings support the development of more accurate susceptibility testing protocols and highlight the need to establish species-specific ECOFF values for dermatophytes.

## 1. Introduction

Over the past decade, our understanding of skin pathogens belonging to the genera *Trichophyton* or *Microsporum* has changed distinctly, particularly regarding their ability to form resistant isolates against antifungal agents. Whereas only few resistant isolates had been identified about 10 years ago [1], and the potential threat was largely underestimated, the treatment of *Trichophyton* infections involving resistant isolates has now become a major challenge for the medical community [2,3,4,5]. Especially, the emergence of a new variant within the *T. mentagrophytes* complex is of concern [6,7,8,9], which has led to widespread superficial dermatophytosis in India and neighboring countries. These infections are frequently resistant to azoles, leading to therapy failure [10,11]. Genome data [12] as well as DNA sequencing data revealed that these isolates represent a distinct *T. mentagrophytes* subtype [9,13], which can be clearly differentiated from *T. interdigitale*. This subtype was later renamed *T. indotineae* [14,15] and represents a basal lineage within the *T. mentagrophytes* complex [12,13]. Nevertheless, there is an ongoing debate about the classification as separated species or as one genotype of *Trichophyton mentagrophytes* complex [16]. A multicenter study conducted across various regions of India revealed a high prevalence of terbinafine-resistant isolates [10]. In fact, *T. indotineae* was identified as the causative agent in the majority of infections, whereas *Trichophyton rubrum* occurrence was reduced to 10% [10]. The high-level resistance against terbinafine is primarily associated with amino acid substitutions in the squalene epoxidase gene *Erg1*, particularly at positions Leu393Phe/Ser and Phe397Leu [10].

Interestingly, *T. indotineae* strains frequently exhibit amino acid exchanges in the *Erg11B* gene [17,18], encoding the sterol 14-α demethylase, an essential enzyme in the ergosterol biosynthesis. Similar to *Aspergillus fumigatus, T. indotinae* possesses two paralogs of *Erg11* (also known as *CYP51*) [19], designated A and B [20]. *Erg11* overexpression leads also to azole resistant phenotypes due to higher enzyme binding capacities for azoles [21,22,23]. Genome analyses showed genomic amplification of a 2.4 kb DNA fragment containing *Erg11B* [21] named type I or the type II amplification of a 7.4 kb DNA fragment of *Erg11B* and two adjacent genes [22]. The high prevalence of multidrug-resistant *T. indotineae* has led to an increase in surveillance of other *Trichophyton* species for antifungal resistance patterns. In India, *T. rubrum* infections also frequently exhibited terbinafine resistance, primarily due to the *Erg1* Phe397Leu point mutation [7]. Increasing numbers of terbinafine-resistant *T. rubrum* isolates have been further reported in Denmark, where various *Erg1* point mutations contributed to the resistance phenotype [24]. Multidrug-resistant *T. rubrum* strains were detected, mostly carrying the *Erg1* Leu393Phe mutation and, in one patient, the *Erg11B* Tyr136His mutation [25]. This latter substitution is well-known among other fungal pathogens and is associated with resistance to short-chain azoles like voriconazole [26]. In one patient isolate, overexpression of multiple-drug-resistant (*MDR*) transporter-like *MDR3* and *MDR4* was observed [25]. A similar *T. rubrum* case report on therapeutic failure with terbinafine, voriconazole, and itraconazole, revealed the uncommon *Erg1* mutations Ile479 Thr and Ile479Val [27]. Additionally, terbinafine-resistant *T. mentagrophytes* isolates from animal sources have been documented [28] as well as itraconazole-resistant *T. quinckeanum* strains identified, belonging to a new ITS subtype [29].

Protocols for determining the inhibitory concentrations (IC values) of antifungal agents have been developed as well as standardized and are summarized in recent literature [30]. The EUCAST E.def9.4 protocol was originally developed for molds [30]. Recently, a specific protocol, the EUCAST E.def11.0, was introduced for *Trichophyton* species [31], which defines breakpoints for terbinafine resistance in *T. indotineae* and *T. rubrum*. A similar CLSI protocol [32] adapted for *Trichophyton* species can also be used for determination of IC values. All of these protocols rely on a defined medium formulation based on the human cell culture medium RPMI1640 with L-glutamine as the primary nitrogen source. Since RPMI1640 is typically used in 5% CO_2_ at 37 °C, the addition of a buffer system is necessary for fungal cultivation under ambient CO_2_ conditions. For this purpose, MOPS buffer adjusted to pH 7.0 is added to RPMI1640 in all susceptibility testing protocols. The original RPMI1640 formulation contains 0.2% (*w*/*v*) glucose, but EUCAST protocols modify this by increasing glucose concentrations up to 2% (*w*/*v*), to better support fungal growth. Nonetheless, some studies reported limited growth of *Trichophyton* isolates in RPMI1640-based media [22]. Leading researchers in some cases revert to the traditional Sabouraud dextrose broth for resistance assays, which contains various peptone sources and supports more robust growth. Historically, the first defined media formulations for *Trichophyton* species, known as *Trichophyton* medium 1–7, were developed based on casein or casamino acids as nitrogen sources, without addition of vitamins [33]. These early formulations underscore the possibility that nitrogen source availability may be a key limiting factor in the growth of *Trichophyton* in RPMI1640-derived media. Itraconazole is a highly lipophilic compound with limited solubility in water [34,35]. The composition of the medium significantly influenced IC values of itraconazole against other fungal pathogens [36].

Resistance genotypes have been validated through corresponding phenotypic resistance measurements. However, the reliability and sensitivity of phenotypic assays still require further optimization. While current protocols provide a valuable framework for correlating specific mutations with resistance profiles, several limitations persist, particularly regarding growth conditions, medium composition, and endpoint interpretation. Inconsistent growth in defined media such as RPMI 1640 may influence inhibitory concentration (IC) determinations and obscure subtle resistance phenotypes. Moreover, variability in inoculum density, incubation time, and reading criteria between laboratories can affect the reproducibility and comparability of results. Therefore, the refinement of phenotypic testing methods remains essential to complement and validate genotypic data accurately.

## 2. Results

### 2.1. Characterization of Dermatophyte Isolates and Overview of Experimental Conditions

For comparative analysis, several strains belonging to the genera *Trichophyton*, *Microsporum,* and *Epidermophyton* were selected and tested using standardized susceptibility protocols, including Eucast E.def11.0 [31], Eucast E.def 9.4 [30], and the CLSI reference method [32]. The complete list of strains used in this study is provided in Table 1. Species identification was confirmed by analyzing the ITS region. GenBank Acc. No. or best sequence matches based on BLAST search results are shown in Appendix A. *T. indotineae* strains from 2017-2018 were collected and analyzed for ITS and *Erg1* in a previous study [10]. Strains from routine cases between 2019 and 2021 were analyzed for ITS, *Erg1*, and *Erg11B* as previously described [17], and the corresponding *T. quinckeanum* genes from both observation periods were also analyzed in an earlier study [29].

The *T. rubrum* strains UKJ 705/21 and UKJ 706/21 were isolated from a patient suffering from tinea corporis generalisata for more than 30 years [27]. Previous terbinafine treatment failed, and only bioavailable itraconazole formulation led to clinical stabilization but not full recovery [25]. Both isolates showed distinct point mutations in the squalene epoxidase gene Erg1, specifically at amino acid position 479 [27] (see Table 1). In addition, sequencing of the sterol 14-α demethylase gene *Erg11B* revealed an amino acid substitution at position Gly443Cys in *T. rubrum* strain UKJ 705/21 (GenBank Acc. No. PV779182) as shown in Table 1. *T. indotineae* isolated from 2017 and 2018 (Table 1) was obtained from a large-scale study on *T. indotineae* isolates collected across various regions of India [10]. Samples from this study were shared with other research groups, who used selected strains for genome analyses [21,22]. Two types of genomic *Erg11B* amplifications were identified [21,22]. For verification of strain identity, type-specific PCR and copy number analysis were performed as previously described [22]. In the future, selected reference strains should be used to improve the comparability of the different antifungal susceptibility testing protocols. Reference strains used in this study, in particular UKJ 1676/17 (TIMM20114), UKJ 1687/17 (TIMM20118), and UKJ 392/18 (TIMM20119), were included in the present analysis as controls representing susceptible (UKJ1676/17) or resistant phenotypes (see Table 1). To classify *Erg11B* amplification types, a specific PCR approach targeting type I or type II fragments was applied as previously described [22] (see Table 1). PCR products were separated via agarose gel electrophoresis and results visualized as shown in Figure 1. The resulting amplification patterns for the three reference strains were consistent with those previously reported for *Erg11B* [22]. However, a notable exception was observed for strain UKJ 1708/17. Although the genome of TIMM20116 (GenBank assembly GCA_023065885.1), like UKJ 1987/17, was classified as type I [22], UKJ 1708/17 exhibited the amplification pattern of a type II strain (see Table 1 and Figure 1b).

The type I primer binding sites are conserved between *T. indotineae* and *T. rubrum,* enabling type-I-specific PCR analysis in both species. However, none of the *T. rubrum* strains tested yielded type I-specific fragments, as shown in Table 1 and Figure 1a.

Additional *T. indotineae* strains listed in Table 1 were obtained from clinical cases at Jena University Hospital and analyzed for both *Erg1* and *Erg11* genotypes and phenotypes [17]. The strains were also included in a recent study examining mRNA expression levels of ergosterol biosynthesis genes, drug efflux transporters and heat shock response genes [23]. One strain, UKJ 476/21, exhibited marked overexpression of *Erg11B* [22] and was further analyzed using both type I and II amplification-specific PCRs. The results confirmed type II genomic amplification of *Erg11B* (see Table 1 and Figure 1b).

Two *T. quinckeanum* isolates, listed in Table 1, were previously found to exhibit different azole resistance profiles [29]. Strain IHEM 13697 was sensitive to itraconazole, while strain UKJ 1506/20 was itraconazole-resistant [29]. Despite the presence of identical *Erg1* and *Erg11B* gene sequences in both strains, several point mutations were identified within *Erg11A* [29], suggesting an alternative mechanism for resistance.

Other strains in the study were derived from clinical submissions, culture collections or control strains used for external quality assessment (RV).

### 2.2. Genomic Copy Number Analysis Reveals Erg11B Amplification Is Specific to T. indotineae Isolates and Absent in T. rubrum

Especially for type II amplification of *Erg11B*, genome assemblies often fail to detect the correct number of genomic copies. Therefore, qPCR assays were employed to determine the copy number of *Erg11B*, as previously described [22], using a similar approach to that used for cDNA expression analysis [23]. Oligonucleotides for *Erg11B* and *Erg11A* were adapted to be suitable for both *T. indotineae* and *T. rubrum* amplification, and all primer sequences are listed in Appendix A. This method enables detection of any type of *Erg11B* genomic amplification. The actin gene was used as a single-copy control, and the mean values of reference strain UKJ 1676/17, harboring single copies for both *Erg11B* and *Erg11A*, were set to one. In all analyzed strains, *Erg11A* consistently behaved as a single copy gene. This is in accordance with genome data analysis of *T. indotineae* strains using BLAST algorithms. The copy number of UKJ 1687/17 matched the previously reported numbers of 5.5 ± 0.6 [22] (see Figure 2). Type I genomes exhibit five to seven genomic copies of *Erg11B* using BLAST algorithms. The type II reference strain UKJ392/18 also showed similar results with 8.9 ± 0.6 estimated copies by qPCR [22] (see Figure 2). In contrast, genome assemblies tended to underestimate *Erg11B* copies in type II isolates, often detecting only one additional copy.

All *T. rubrum* strains analyzed showed a single-copy configuration of *Erg11B* (Figure 2), indicating that to date, genomic amplification of this gene is restricted to *T. indotineae*.

### 2.3. Influence of Growth Medium and Temperature on Phenotypic Discrimination Azole-Resistant and Sensitive Strains by Itraconazole Inhibitory Concentration (IC) Measurements

Measurements of inhibitory concentrations (ICs) depend on several standardized protocols, including EUCAST E.def.11, EUCAST E.def.9.4 or CLSI [30,31,32]. All three protocols are based on the defined cell culture medium RPMI1640 (abbreviated as RP) as growth medium with MOPS as a buffer component to maintain pH stability. In EUCAST protocols, the basal glucose concentration of 0.2% (*w*/*v*) in RP is increased to 2% (*w*/*v*), and this modified medium is referred to as RG for better clarity here. However, several authors reported that RP or RG is not optimal for growth of dermatomycetes [22], and our results provide further evidence supporting this observation. Especially, species belonging to the genus *Microsporum* exhibited severely impaired growth in RG medium, which made IC determination impossible (Appendix A). To address this limitation, a new medium variant was developed, drawing on knowledge for defined media formulations for *Trichophyton* species [33]. While the nitrogen source in RP medium is the amino acid L-glutamine, classical *Trichophyton* media rely on casein or casamino acids, free of vitamins [33]. Based on this, RG medium was supplemented with 2.5 g/L casein hydrolysate (vitamin-free, Carl Roth, Karlsruhe, Germany), and the resulting medium was designated as RGC. In addition, Sabouraud-glucose broth (SG, Merck, Darmstadt, Germany) was used as control medium representing optimal growth conditions for dermatophyte isolates.

Another critical variable among the three protocols for determining ICs is the range of incubation temperatures. EUCAST E.def.11 stipulates incubation at 25–28 °C, whereas both EUCAST E.def.9.4 and the CLSI protocol employ higher temperatures of 37 °C ± 3 °C. To investigate the impact of growth temperature and medium composition on IC values, two incubation temperatures, 28 °C and 34 °C, were selected, as along with three growth media, RG, RGC, and SG. All strains listed in Table 1 were tested under these conditions. Spore concentrations followed the EUCAST E.def.11 protocol. For a subset of strains, additional IC values were obtained using the CLSI protocol, which utilizes lower spore concentrations in combination with RP medium. All measurements were performed using microplate-laser-nephelometry (MLN) as previously described [29]. Two microtiter plates were processed in parallel; one plate was continuously shaken and automatically measured every hour for a total duration of 120 h. The second plate was incubated without shaking and measured manually at 72, 96, and 120 h. IC_50_ values were calculated using the logistic regression model of the program OriginPro 2025. All resulting values are listed in Appendix A. At the 72 h time point, data were frequently incomplete due to extended lag phases in the growth curves of certain isolates, preventing accurate IC determination at this time point (Appendix A). Even at 96 h, IC values could not be determined for some slow-growing isolates that only began exponential growth at later time points (Appendix A).

Based on their genotype listed in Table 1, the *T. indotineae* strains UKJ 1687/17, UKJ 1708/17, UKJ 392/18, and UKJ 476/21 represent azole-resistant strains, whereas UKJ 1676/17, UKJ262/21, CBS 146726, and CBS 146727 belong to the azole-sensitive group. IC_50_ values of these two groups differed consistently, regardless of the growth medium used, as illustrated in Figure 3. Nevertheless, statistical significance between results for resistant and sensitive groups markedly increased when SG medium was used for cultivation. The highest absolute IC_50_ values were observed for RGC medium (Appendix A), which also supported the most robust growth of the isolates.

An increase in growth temperature consistently led to decreased IC_50_ values, regardless of the growth medium used. This indicates that temperature also has a significant influence on determination of IC values. Both *T. rubrum* strains, UKJ705/21 und UKJ706/21, exhibited elevated IC_50_ values across all tested media (Appendix A) compared to DSM16111. Likewise, differences in azole susceptibility between the two *T. quinckeanum* strains were confirmed, as previously reported [29], independent of the growth medium applied. The datasets of *T. rubrum*, *T. indotineae,* and *T. quinckeanum* isolates were then combined for comparative analysis. As observed earlier, the use of SG medium provided the clearest separation between resistant and sensitive isolates, with no overlap in concentration ranges (Appendix A). This underscores the utility of SG as the most suitable medium for defining a reliable threshold (cutoff value) between azole-sensitive and -resistant isolates. While IC_50_ values obtained using RG and RGC media also allow differentiation between sensitive and resistant isolates, the separation was less distinct, and overlapping IC ranges between both groups were observed (Figure 3 and Appendix A).

### 2.4. Temperature Shift to 34 °C Resulted in Reduced IC Values Across All Media, with the Strongest Effect Observed in SG Medium

Results from other isolates revealed varying levels of IC_50_ values among both *T. tonsurans* strains. Notably, strain UKJ186/23 demonstrated the highest sensitivity to itraconazole (Appendix A). Strains belonging to the *T. benhamiae*/*erinacei* complex showed IC_50_ values comparable to or even exceeding those of the resistant *T. indotineae* strains (Appendix A), suggesting a species-specific intrinsic tolerance to itraconazole. For the *M. canis* strains, IC_50_ values were strongly influences by temperature. An increase to 34 °C resulted in a roughly tenfold decrease in IC values. Interestingly, both *M. canis* strains were able to grow at 28 °C using RG but failed to grow at 34 °C in the same medium (Appendix A). These findings point to a heat-dependent sensitivity and suggest that the optimal growth temperature for *M. canis* is lower than that of *Trichophyton* species. Interestingly, the medium-dependent effect on IC values, especially the SG-related increase in separation between resistant and sensitive strains, was less pronounced for *T. interdigitale* strains (Appendix A).

The complete 120 h dataset of IC values from *Trichophyton* strains clearly demonstrated the overall influence of temperature and medium composition on antifungal susceptibility (Figure 4). Interestingly, the combination of 34 °C and SG medium consistently resulted in lower IC value range, suggesting that these conditions maximize the antifungal activity of itraconazole.

### 2.5. RGC Medium Enhances Growth Compared to RG and Shows Intermediate Performance Between RG and SG Medium

The addition of casein as nitrogen source modified fungal growth, shifting the pattern closer to that observed in SG medium and thereby overcoming some of the limitations with RG medium. Growth curves at 28 °C in RG medium exhibited a shortened exponential growth phase, followed by a marked reduction in growth rate and difficulties in reaching the plateau phase within the 120 h observation window. At the end of this period, only a thin mycelial mat had formed, in stark contrast to the much thicker and more robust mycelial mats seen in SG or RGC media. Continuously hourly measurement throughout the fungal growth phase enabled determination of growth parameters such as doubling time and the time to reach the plateau phase, using Origin2025. Growth in RG at 28 °C resulted in a significantly prolonged doubling time for all tested fungal strains (Figure 5a). At 34 °C, the doubling times in RG remained extended when compared to SG, whereas values for RGC and RG were more similar (Figure 5a). The differences in growth behavior became even more evident when examining the time required for reaching the plateau phase (Figure 5b). Dermatophytes grown in RG showed a significant delay in reaching this phase (Figure 5b). At 28 °C, most strains required nearly the full 120 h to approach the plateau. Since the observation window ended at 120 h, it is possible that some strains would have eventually completed the growth cycle given more time, and therefore the distribution of plateau-phase times for RG at 28 °C was consequently compressed (Figure 5b).

### 2.6. Species-Specific Growth Analysis Revealed Impaired Growth of T. benhamiae/erinacei and T. indotineae in RG Medium

Species-dependent analysis of growth behavior showed notable differences in both doubling time and the time required for reaching the plateau phase.

*T. indotineae* strains demonstrated shorter doubling times in RGC medium compared to SG medium (Figure 6a), indicating that the addition of casein sufficiently improved RG medium to support optimal growth in this species. In contrast, strains of the *T. benhamiae* complex exhibited the shortest doubling times in SG medium, while results for RGC and RG were more similar (Figure 6a). This suggests that both RG and RGC medium are suboptimal for the growth of these species. For *T. rubrum*, *T. interdigitale,* and *T. mentagrophytes,* no significant differences in doubling times were observed across the tested growth media (Figure 6a), suggesting that these species are less sensitive to the specific nitrogen source or medium composition. Analysis of the time required for reaching the plateau phase revealed that, across all tested species, RG medium resulted in the longest periods, while SG medium yielded the shortest (Figure 6b). This underscores SG medium as the most favorable for promoting rapid and robust fungal growth for nearly all strains. When using plateau phase time as an indicator of growth efficiency, *T. indotineae* showed improved performance in RGC medium, a pattern that was also observed for *T. interdigitale*, *T. mentagrophytes,* and the *T. benhamiae* complex (Figure 6b). No significant effect was found for *T. rubrum* strains (Figure 6b), indicating that for this species the RG medium is less problematic.

### 2.7. Effect of Spore Titer Reduction on LAG Phase Duration Revealed Impaired Growth in RG Medium

The slow growth observed in RG medium at low temperatures may explain why the EUCAST E.def11.0 protocol relies on high conidia concentrations, typically between 2 × 10^5^ and 10^6^ cfu/mL. For the majority of strains, achieving this conidia density was not problematic when using Dermasel agar (Thermoscientific, Wesel, Germany) for pre-cultivation of isolates. However, some strains produced markedly lower amounts of conidia, making it difficult to reach the required spore concentration for standardized inoculation. Interestingly, the CLSI protocol allows for distinctly lower conidia concentrations (1–3 × 10^3^ cfu/mL), making it more adaptable to a broader range of isolates, particularly those with poor sporulation. The high spore density required by the EUCAST methods also results in an elevated background turbidity, typically 10–20%, as measured by MLN. This increased background likely results from cell debris, which may release an undefined mixture of proteins, amino acids, and vitamins into the medium, thereby altering its composition in unpredictable ways.

Analysis of lag phases in the growth controls for all strains used in the IC_50_ determination of itraconazole revealed largely similar durations, regardless of growth medium or temperature (Appendix A). However, when strains struggled to produce sufficient amounts of spores and targeted inoculum density could not be achieved, prolonged lag phases were occasionally observed. To assess the influence of spore concentrations on the length of the lag phase, serial tenfold dilutions of the spore suspensions were analyzed across all media types (RGC, RG, and SG) using MLN. Due to their poor sporulation, all *Microsporum* and *Epidermophyton* strains were excluded from this analysis. A total of 20 *Trichophyton* strains were analyzed, with exclusion of UKJ186/23, DSM16111, DSM16110, and IHEM13697. As expected, reduced spore concentrations consistently led to prolonged lag phases (Figure 7), and each of the three tested dilution levels showed significant differences independent of medium and temperature (Appendix A). Further, significant differences were observed when comparing RGC and RG at 28 °C (Figure 7). Analysis of lag phases across all strains with high spore titers showed significant differences only between RG and SG at 28 °C (Appendix A). In contrast, reduced spore concentrations caused a marked increase in lag phase that was strongly dependent on the medium used (Figure 7). The most pronounced delay was observed in RG medium, where the lowest spore titer resulted in significantly extended lag phases compared to RGC and SG at both tested temperatures (Figure 7). This effect was especially evident in strains of *T. benhamiae var. luteum,* which exhibited severe growth delays in RG when spore numbers were limited, accounting for the outliers in Figure 7.

## 3. Discussion

Phenotypic analysis should provide sufficient resolution to distinguish between isolate-specific genotypes of genes involved in antifungal resistance mechanisms. However, a drawback of the EUCAST E.def 11.0 and CLSI methods is their reliance on RG or RP medium, variants of RPMI1640. These cell culture media are suboptimal for dermatophyte growth, particular for *Microsporum* strains, which showed severe growth impairments. Even *Trichophyton benhamiae* showed compromised growth under these conditions. The addition of casein improved growth performance in RGC medium to some extent but did not fully match the growth quality achieved with SG medium. This highlights the importance of complex nitrogen sources for optimal dermatophyte growth. Both *Trichophyton* and *Microsporum* species are highly adapted to keratin utilization, as reflected by their extensive repertoire of endo- and exoproteases [38]. Within their ecological niches, they encounter little evolutionary pressure to utilize inorganic nitrogen sources such as nitrate or ammonium salts. RPMI1640 is based on high amounts of arginine and glutamine, whereas most other amino acids are underrepresented. This imbalance likely does not meet the metabolic requirements of dermatophytes; hence, the synthetic *Trichophyton* media were developed based on casein or casamino acids as a nitrogen source [33], indicating the necessity of complex organic nitrogen sources for adequate fungal growth. Therefore, standard susceptibility testing protocols might underestimate resistance phenotypes in some dermatophytes due to insufficient growth under test conditions. For the differentiation between resistant and sensitive isolates, SG medium proved to be the most suitable growth medium. Itraconazole IC_50_ values obtained for *T. indotineae* as well as from *T. rubrum* and *T. quinckeanum* isolates grown in SG medium showed a clear separation between resistant and sensitive strains (Figure 3 and Appendix A), corresponding with their respective genotypes. These results support findings from other authors who also recommended SG medium for routine cultivation of dermatophytes [22,39]. Moreover, SG is commercially available, can be autoclaved, and is thus easy to handle in standard laboratory settings. The reason why RG as well as RGC medium resulted in generally higher IC_50_ values and showed overlap between resistant and sensitive isolates, in contrast to SG, remains unclear. Itraconazole is a lipophilic compound with poor solubility in aqueous solutions [34,35], and its solubility is further dependent on the pH of the medium [40,41,42]. Acidic environments increase the solubility of itraconazole [40,41]. RG and RGC are buffered with MOPS at pH 7, whereas SG medium lacks buffering components and, therefore, has limited buffer capacity, possibly enhancing solubility of itraconazole during cultivation. Protein degradation in SG medium may mimic early infections stages more closely, as *Trichophyton* species naturally grow on keratinized tissue under mildly acidic skin conditions (pH 4–5) [38]. Such pH shifts are known to influence gene expression in *Trichophyton rubrum,* including genes encoding specialized permeases and proteases [43]. Permeases could be involved in itraconazole uptake, which may explain the decreased IC values observed in SG medium. In accordance, SG possibly provides more physiologically relevant conditions for antifungal susceptibility testing of dermatophytes.

The high spore concentration required by EUCAST protocols has two major drawbacks. First, it may mask growth deficiencies in certain isolates, and second, cell debris from high spore loads can release undefined proteins and metabolites that unintentionally influence growth. As a result, even within one species, reproducibility is reduced if different spore production capacities exist among isolates. Isolates that produce only low amounts of spores had to be excluded from the analysis, which limits the applicability of the EUCAST protocol across broader strain diversity. In this respect, the CSLI protocol allows for lower spore concentrations, offering an advantage compared to EUCAST protocols in terms of broader usability and reducing the influence of background materials. Furthermore, higher incubation temperatures accelerated growth speed, improving time efficiency of the assay. Comparison of the obtained results with previously published IC values for *T. indotineae* UKJ1676/17 and UKJ1687/17, 0.06 and 0.5 µg/mL for itraconazole, respectively [22], showed high consistency with our findings. In our study, IC_50_ values for UKJ1687/17 were 0.294 µg/mL at 28 °C and 0.29 µg/mL at 34 °C, whereas UKJ1676/17 showed values of 0.028 µg/mL and 0.03 µg/mL, respectively with high spore concentrations and SG as growth medium obtained over 120 h (Appendix A). This 10-fold difference of resistant and sensitive isolates corresponds closely with the recently obtained data using the CLSI conditions and SG as medium [22]. Interestingly, when using drastically reduced spore concentration (10^3^ cfu/mL), about 100-fold lower, and CLSI conditions with RP medium, the difference between resistant and sensitive isolates increased substantially, albeit overall lower azole amounts were needed. Under these conditions at 34 °C, UKJ 1687/17 showed an IC_50_ value of 0.088 µg/mL, while UKJ 1676/17 featured one of only 0.0008 µg/mL (Appendix A). A comparable trend was observed for *T. quinckeanum* isolates under these conditions. The resistant *T. quinckeanum* UKJ 1506/20 showed an IC_90_ of 0.34 µg/mL and the sensitive strain IHEM13697 of 0.003 µg/mL [29]. When tested with high spore concentrations (EUCAST conditions), SG as growth medium and 120 h incubation, UKJ1506/20 showed IC_50_ values of 0.39 at 28 °C and 0.19 µg/mL at 34 °C, whereas IHEM13697 showed 0.026 and 0.009 µg/mL, respectively (Appendix A). This corresponds to a 15-fold (28 °C) and 21-fold (34 °C) difference in IC values between sensitive and resistant isolates. This further highlights the influence of spore concentrations on assay resolution and indicates usage of lower spore concentrations for IC determination. These results demonstrate that lower spore concentrations improve the dynamic range of IC values and facilitate the identification of genotypic differences with smaller phenotypic impacts. At the same time, this approach exposes the limitations of insufficient media formulations such as RG or RP for certain *Trichophyton* and *Microsporum* species. Therefore, when using low inoculum levels, the growth medium must be fully adequate in its composition. The reduced influence of background materials under these conditions allows more reliable differentiation of isolate-specific antifungal susceptibility profiles.

Yet, the phenotypic expression of Gly443 mutations can be difficult to interpret [17,18], although comparable mutations have been identified in several fungal pathogens [26,44,45] and protein modeling has highlighted the importance of this position for itraconazole binding [46]. Other resistance mechanisms may overlap with or mask the effects of *Erg11B* amino acid changes; for example, the overexpression of efflux transporters [23,47,48]. Another such mechanism is the overexpression of *Erg11B* itself, which can lead to elevated levels of the encoded sterol 14-α demethylase. This may result in the sequestration of azoles through binding, thereby reducing the drug’s inhibitory effect. Genomic amplification of *Erg11B* is a key contributor leading to increased mRNA and resulting protein levels [21]. Two distinct types of genomic *Erg11B* amplification have been identified [22]. Type I amplification is based on five to seven copies of a 2.4 kb fragment containing the *Erg11B* gene, sequentially arranged in the same orientation [22]. Interestingly, all amplified copies in type I isolates carry the *Erg11B* Gly443Glu mutation, suggesting that the amplification originated from an *Erg11B* mutant strain. In contrast, type II isolates harbor an amplification of a 7.4 kb fragment encompassing *Erg11B* along with two adjacent genes located upstream and downstream of *Erg11B* [22]. Type II isolates represent the more prevalent subpopulation of *T. indotineae* [22] and carry the wild-type *Erg11B* sequence. Interestingly, type II isolates also frequently harbor the *Erg1* Ala448Thr mutation [22]. While genome sequencing and quantitative PCR (qPCR) data yielded similar results for type I [22], the copy number and presence of amplification events appear to be underestimated in genomic assemblies for type II [22]. This limitation persists despite the availability of very high-quality genome assembly data [22] with scaffold numbers approaching the expected chromosome numbers for *Trichophyton* species [49]. A particularly well-resolved genome assembly was generated from strain TIMM 20114 (UKJ1676/17), which serves as a reference for a single-copy *Erg11B* gene. Nevertheless, discrepancies between genomic copy number and qPCR results remain for certain isolates when single-copy genes are used as internal controls [22]. The methods developed to distinguish between type II and I genomic *Erg11B* amplifications were adapted and subsequently applied to analyze clinical isolates from the UKJ strain collection. Overexpression of *Erg11B* at the mRNA level has been consistently associated with increased azole resistance [21,22,23]. Interestingly, environmental stress conditions, such as constant exposure to elevated temperatures (e.g., 37 °C) induced a heat stress response that led to an increase in *Erg11B* expression in the sensitive *T. indotineae* strain [23]. Remarkably, high *Erg11B* expression levels were detected even when *Erg1* and *Erg11A* were downregulated under the same conditions [23], suggesting functional divergence between the two *Erg11* paralogs. The co-regulation of *Erg1* and *Erg11A* indicate their central role in ergosterol biosynthesis, whereas *Erg11B* appears to function as a stress-responsive gene. Interestingly, the genomic co-amplification of the genes with *Erg11B* in type II isolates are involved in DNA repair and telomere maintenance [23], indicating a possible stress adaption linked to *Erg11B*. In *A. fumigatus*, maintaining eburicol homeostasis within the cell is critical, as intracellular accumulation of this sterol intermediate can lead to toxic effects and cell death [50]. Therefore, *T. indotineae* Erg11B may exert a protective function, helping to regulate eburicol levels and prevent toxic concentrations.

The strong correlation between genotype and phenotype among *T. indotineae*, *T. rubrum*, and *T. quinckeanum* enables the establishment of epidemiological cutoff values (ECOFFs) to distinguish resistant from sensitive isolates. This phenotypic consistency supports the use of standardized IC_50_ values or simplified agar-based plate assays, as recently described [29], for rapid and cost-effective screening in diagnostic laboratories. The genomic amplification of *Erg11B* leads to azole resistance in *T. indotineae*; in accordance, application of type II and I specific PCRs could offer a valuable molecular diagnostic tool, particularly in regions where azole-resistant strains of *T. indotineae* are emerging. These PCR-based methods provide a reliable and straightforward approach to identifying known amplification types, aiding treatment decisions and epidemiological tracking. The consistent co-occurrence of type II *Erg11B* amplification and the *Erg1* Ala448Thr mutation further suggests that these strains represent a subpopulation of clonal lineage within *T. indotineae*. This lineage likely emerged through a limited number of recombination or duplication events and has subsequently spread, as reflected by its frequent detection in epidemiological studies. Such genomic signatures could be used to trace the origin and dissemination of resistant subpopulations and provide insight into the evolutionary dynamics of azole resistance in dermatophytes.

In conclusion, a combination of phenotypic testing under optimized growth conditions and targeted molecular diagnostics enables a robust and comprehensive resistance surveillance system. This dual approach not only enhances diagnostic accuracy but also helps to identify emerging resistance mechanisms and guide appropriate antifungal therapy.

## 4. Materials and Methods

### 4.1. Strains and Growth Conditions

Strains were obtained from routine cases at Jena University Hospital (UKJ) or studies and strain collections as listed in Table 1. Strains were pre-cultivated for three to eight weeks for MLN analysis. The genotypes of strains were analyzed by sequencing of ITS regions and results including GenBank Acc.No. and best BLAST hits are listed in Appendix A. For the media RG, RP and RGC RPMI1640 containing L-glutamine (Capricorn Scientific, Ebsdorfergrund, Germany) were the basis of the media variants. Stock solutions of 1.5 M MOPS/NaOH pH 7.0, 30% *w*/*v* glucose, and 25 g/L casein hydrolysate (Carl Roth, Karlsruhe, Germany) were used to prepare the final media. RG, RP, and RGC contained final concentration of 150 mM MOPS/NaOH pH 7.0 by dilutions of stocks. RG and RGC contained final concentration of 2% *w*/*v* glucose from stock solution considering the fact that RPMI1640 itself contained 0.2% *w*/*v* glucose. For RGC, casein stock was diluted to final concentration of 2.5 g/L. SG was prepared using Sabouraud-2% *w*/*v* glucose bouillon (Merck, Darmstadt, Germany).

### 4.2. Type II and I Specific PCR, qPCR, and DNA Sequencing

ITS PCR primers and conditions were used as described [9] and eluted PCR fragments were sequenced. *Erg11B* of *T. rubrum* was sequenced as described [17] using *T. indotineae* primers that were conserved between both species. For identification of genomic amplification type I fragments of *Erg11B*, PCR was performed using primers as described [22] listed in Appendix A. The primers were derived from conserved positions and were suitable for *T. indotineae* as well as for *T. rubrum*. PCR conditions were as described with annealing temperature of 56 °C [9]. Fragments were separated using agarose gel electrophoresis and stained with ethidium-bromide solution. PCR conditions for genomic type II amplification were as described for qPCR [23]. Primer sequences were as described [22] and listed in Appendix A. The amplified fragments were separated with agarose gel electrophoresis.

*Erg11B* copy numbers were determined with qPCR conditions as described for cDNA analysis [23] using genomic DNA as source. Primers were adapted that could be used for amplification of *T. indotineae* as well as of *T. rubrum* DNA and listed in Appendix A. Actin qPCR primers were as described [47] and listed in Appendix A. Genomic DNA of UKJ 1676/17 served as *Erg11B* single copy control due to genome information listed in Table 1. *Actin* and *Erg11A* fragments showed single copy status for all strains. To display the copy number of *Erg11B* of all *T. indotineae* and *T. rubrum* isolates, the CT values of RT_PCR of *Erg11B* were normalized against *Act1* gene and secondly against *Erg11A* using the 2^−ΔΔCT^ method [37]. The algorithm was corrected for primer efficiency (1.9 instead of 2) as described [51]. Mean values of isolate UKJ1676/17 were normalized as one-copy genes and values of all other isolates were set in relation to UKJ1676/17. Two biological replicates and two technical replicates were determined with qPCR.

### 4.3. Microplate Laser Nephelometry (MLN) Assays

MLN conditions were as previously described [29]. The spore suspensions were adjusted to final concentrations of 2 × 10^5^ cfu/mL for all experiments using RG, RGC, and SG as medium. For experiments using RP medium, the spore concentrations were reduced to 2 × 10^3^ cfu/mL. Two plates were always prepared in parallel. One plate was measured every hour for a 120 h period. The second plate was stored in an incubator and measured with MLN immediately after preparation at 1 h, 72 h, 96 h, and 120 h manually. For determination of growth curves with MLN, final spore concentrations were analyzed of 2 × 10^5^, 2 × 10^4^ and 2 × 10^3^ cfu/mL, respectively. The plates were measured every hour for 120 h period.

### 4.4. Graphical Image Preparation, Data Performance, and Statistical Analysis

Graphical images were prepared using the OriginPro 2025 (Origin-Lab Corporation, Northampton, MA, USA) software. IC50 values were performed for non-linear curves using logistic option with Levenberg Marquardt algorithm of OriginPro2025. The following term was used: y = A2 + (A1 − A2)/(1 + (x/x0)^p). Initial value (A1) and final value (A2) were used as predicted fixed values, followed by using the fit option. The center of the curve (x0) represented the IC50 value. Growth curves were analyzed for the exponential phase using non-linear curves with the ExpGrow1 function of OriginPro2025. The following term was used: y = y0 + A1 × exp ((x − x0)/t1). The values y0 and x0 were used as predicted fixed values and optimized using the fit function. Time-doubling values were obtained as produced from output data t1 × ln(2) factor (0.693147181). Lag and plateau phases were determined manually from growth curves. Statistical analysis was performed with IBM SPSS Statistics 30. A pairwise Mann–Whitney U test was used to determine the asymptotic significance values.

## Figures and Tables

**Figure 1 ijms-26-07090-f001:**
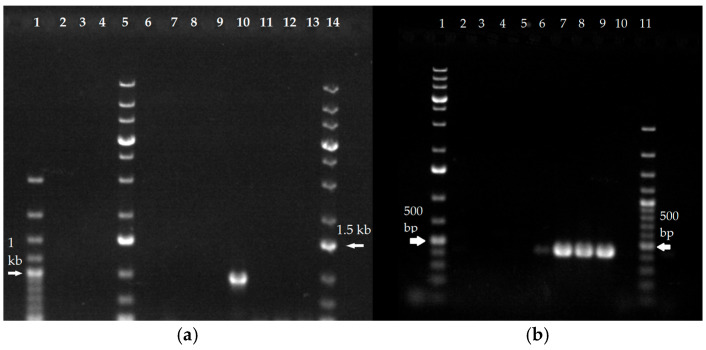
Genomic amplification type-specific PCR of *T. indotineae* and *T. rubrum* isolates confirmed genome analysis of TIMM strains. *Erg11B* genomic amplification type I specific PCR (**a**) and type II (**b**). PCR products were separated via agarose-gel-electrophoresis. *T. rubrum* strains DSM16111, UKJ705/21, UKJ 706/21 are shown in lines 2–4 (**a**); azole-sensitive *T. indotineae* strains UKJ1676/17, CBS146726, CBS146727, UKJ262/21 are in lines 6–9 (**a**) and lines 2–5 (**b**); azole-resistant *T. indotineae* strains UKJ1687/17, UKJ1708/17; UKJ392/18, UKJ476/21 are in lines 10–13 (**a**) and lines 6–9 (**b**). A 100 bp-plus DNA ladder was used as size marker in line 1 (**a**) and line 11 (**b**) and a 1 kb-plus DNA ladder was used in lines 5 and 14 (**a**) and line 1 (**b**).

**Figure 2 ijms-26-07090-f002:**
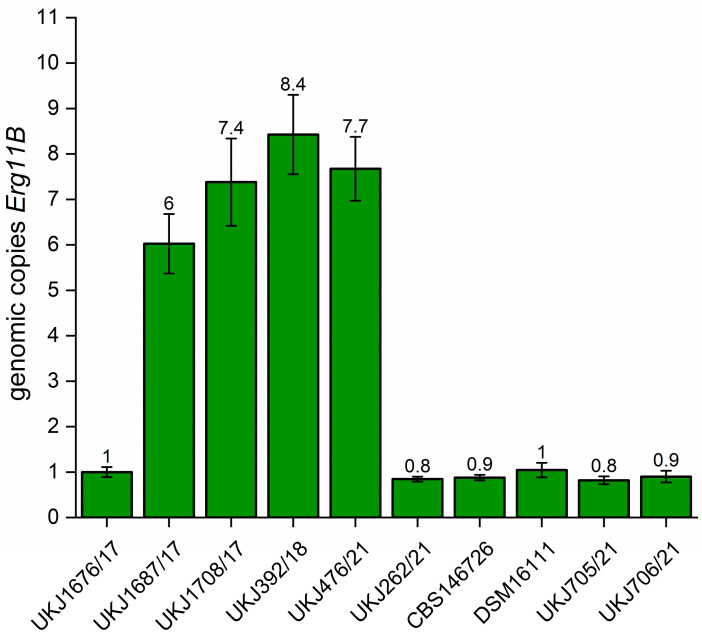
Quantification of *Erg11B* copy number demonstrates genomic amplification in *T. indotineae,* while all *T. rubrum* isolates maintain a single-copy status. The genomic copy number of *Erg11B* was determined by quantitative real-time PCR (qPCR) using genomic DNA as template, as described recently [22]. Primers were designed and optimized for *T. indotineae* as well as for *T. rubrum* and are listed in Appendix A. Strain UKJ1676/17 (TIMM20114), known to harbor a single copy of Erg11B, was used as reference and its mean value was normalized to one copy. Two single-copy reference genes, actin and *Erg11A*, were used to calculate the relative fold change of *Erg11B* employing the 2^−ΔΔCT^ method [37].

**Figure 3 ijms-26-07090-f003:**
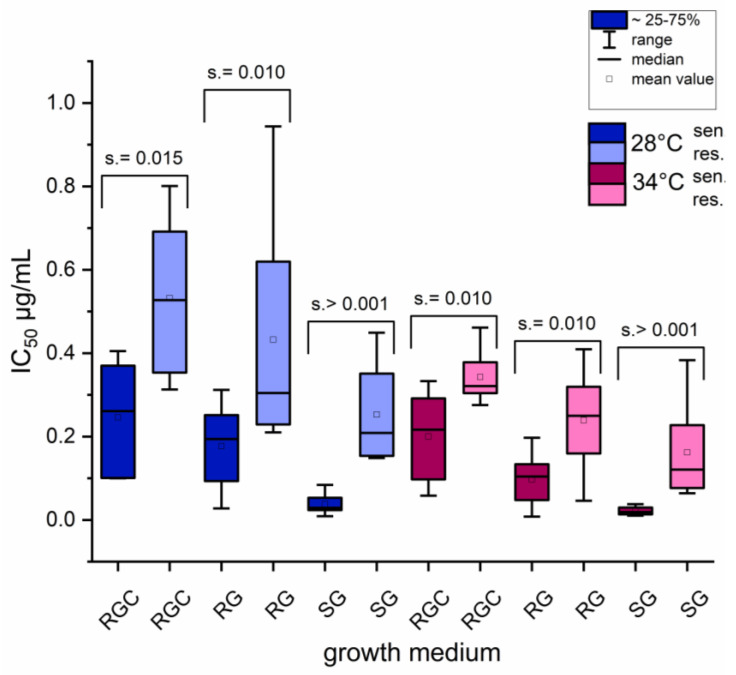
Significant discrimination of itraconazole IC_50_ values between sensitive and resistant *T. indotineae* isolates using SG growth medium. Itraconazole IC_50_ values after 120 h in different growth media (RGC, RG, and SG) obtained for sensitive and resistant *T. indotineae* isolates. Resistant strains (UKJ 1687/17, UKJ 1708/17, UKJ 382/18, and UKJ 476/21) and sensitive strains (UKJ 1676/17, UKJ 594/19, UKJ 1145/19, and UKJ 262/21) were grouped according to their genotypes presented in Table 1 and previous measurements [17,22]. Statistical analysis was performed using pairwise Mann–Whitney U tests; asymptotic significance values are indicated.

**Figure 4 ijms-26-07090-f004:**
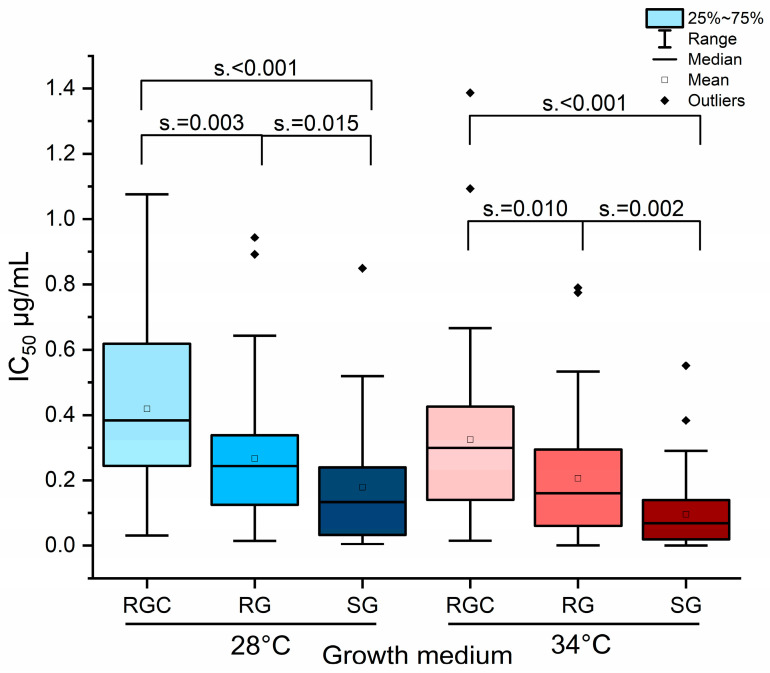
IC values of all fungal isolates showed of the strongest itraconazole effect at 34 °C in SG medium. Comparison of itraconazole IC_50_ values for all isolates obtained after 120 h in the different growth media (RGC, RG, and SG) and at the two cultivation temperatures. Data were analyzed using pairwise Mann–Whitney U tests. Asymptotic values for significance (s.) are indicated to illustrate statistically relevant differences between conditions.

**Figure 5 ijms-26-07090-f005:**
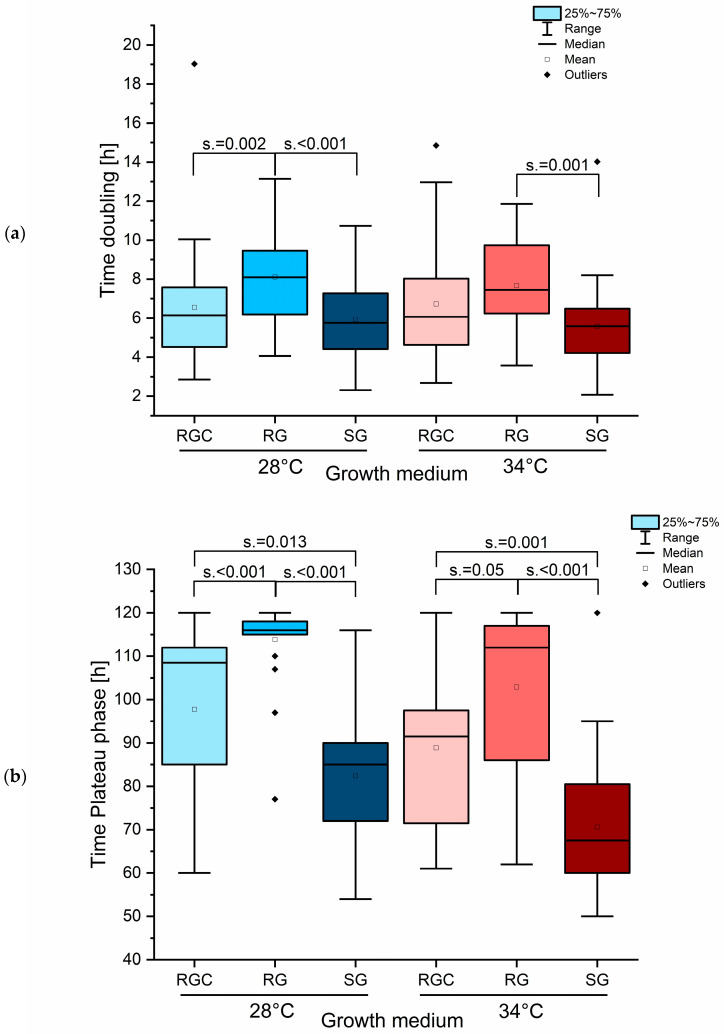
Significant increase of doubling time (**a**) and time to reach plateau phase (**b**) observed in RG medium compared to SG and RGC. Growth media RGC, RG, and SG influenced growth parameters such as doubling time and time to reaching plateau phase across fungal strains. Cultures were monitored over 120 h. Pairwise comparisons using Mann–Whitney U tests revealed significant differences between media. Asymptotic values for significance (s.) are indicated.

**Figure 6 ijms-26-07090-f006:**
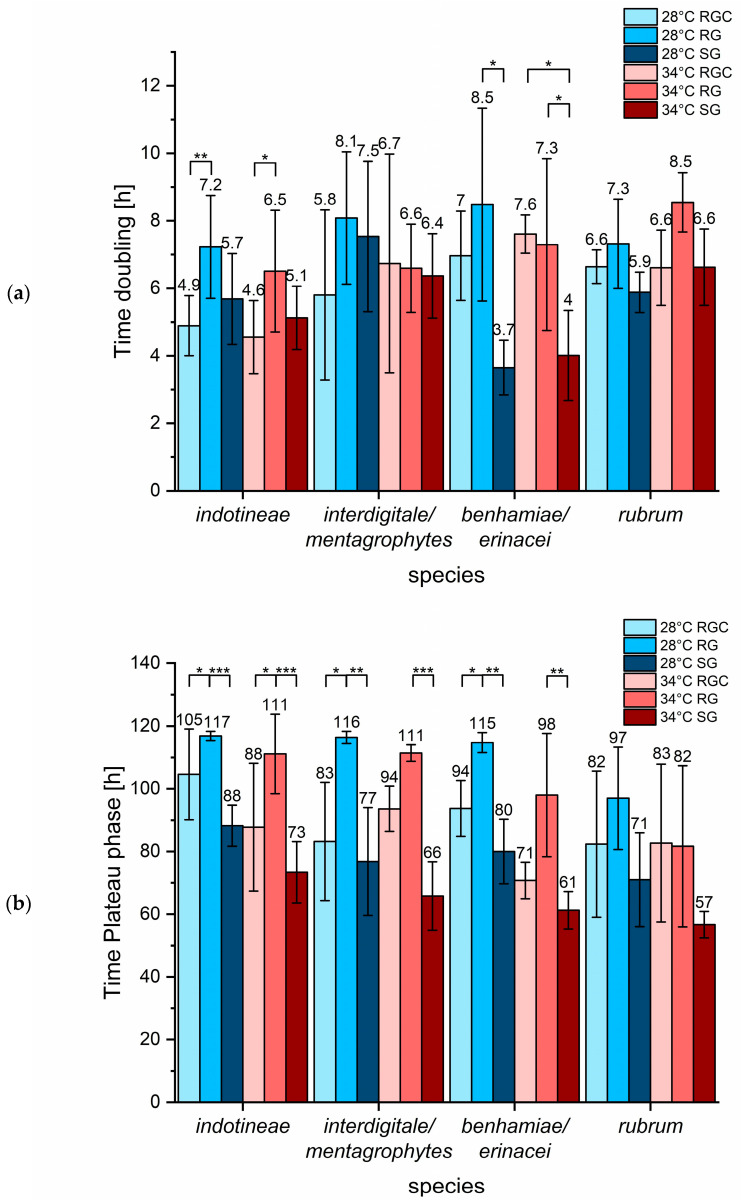
Species-dependent increase of doubling time (**a**) and delayed plateau phase (**b**) in RG medium observed for *T. indotineae* and the *T. benhamiae* complex. Growth parameters were analyzed for *T. indotineae* (*n* = 8), *T. interdigitale*/*mentagrophytes* (*n* = 5), *T. benhamiae* complex (*n* = 4), and *T. rubrum* (*n* = 3) strains in different media over 120 h. Doubling time (**a**) and the time to reach plateau phase (**b**) are shown. Mean values are indicated above each column. Statistical analysis was performed using pairwise Mann–Whitney U tests. Asymptotic significance levels (s.) are indicated (* s. < 0.05; ** s. < 0.01; *** s. < 0.001).

**Figure 7 ijms-26-07090-f007:**
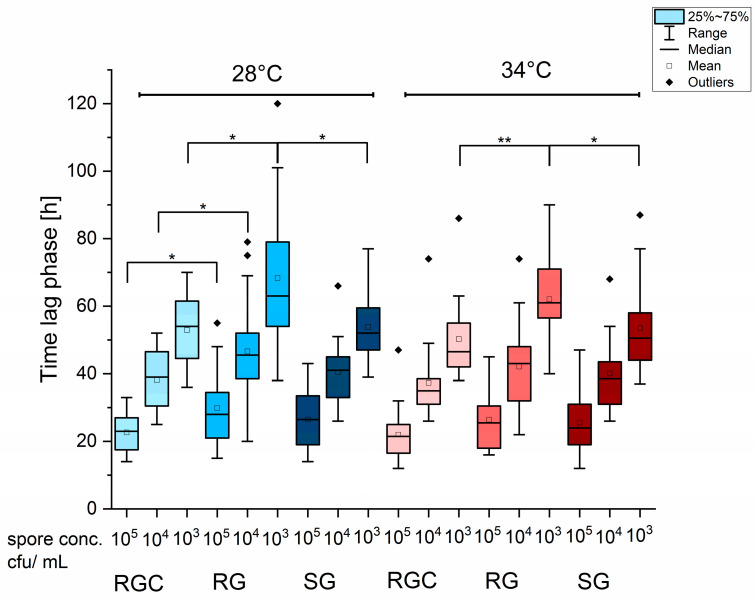
Dilution of spore titers prolongs the lag phase in *Trichophyton* strains. Lag phases were measured for decreasing spore titers (10^5^ cfu/mL, 10^4^ cfu/mL, and 10^3^ cfu/mL) in 20 *Trichophyton strains*. Significant differences were consistently seen between spore concentrations (see Appendix A). Within each spore titer, significant differences between media are indicated by brackets with stars. Data were analyzed using pairwise Mann–Whitney U tests. Asymptotic significance values e (s.) are indicated (* s. < 0.05; ** s. < 0.01).

**Table 1 ijms-26-07090-t001:** Overview of strains, and their associated *Erg1* and *Erg11B* genotypes.

Genus	Species	ITS Subtype, Variety	Collection No.	Collection Syn.	*Erg1* Amino Acid Exchanges	*Erg11B* Amino Acid Exchanges	*Erg11B* Amplification Type	Source, Cited
*Trichophyton*	*rubrum*		_UKJ_705/21	202247/21	Ile479Thr	Gly443Cys	No amplificates	[27]
			_UKJ_706/21	202252/21	Ile479Val	Wild-type	No amplificates	[27]
			_DSM_16111		Wild-type	Wild-type	No amplificates	DSMZ
	*tonsurans*		_UKJ_186/23		n.d.	n.d.	n.d.	Routine
			_UKJ_1163/23		n.d.	n.d.	n.d.	RV491/23
	*interdigitale*	I	_DSM_16110		n.d.	n.d.	n.d.	DSMZ
		II	_DSM_4167		n.d.	n.d.	n.d.	DSMZ
		II	_UKJ_1780/22		n.d.	n.d.	n.d.	Routine
	*indotineae*	VIII	_UKJ_1676/17	_TIMM_20114	Ala448Thr	Wild-type	No amplificates	[10,21]
		VIII	_UKJ_1687/17	_TIMM_20118	Phe397Leu	Gly443Glu	Type I	[10,21]
		VIII	_UKJ_1708/17		Ala448Thr	Wild-type	Type II	[10,21]
		VIII	_UKJ_392/18	_TIMM_20117, 200087/18	Ala448Thr	Wild-type	Type II	[10,21]
		VIII	_CBS_146726	_UKJ_594/19	Phe397Leu	Tyr444His	No amplificates	[17,23]
		VIII	_CBS_146727	_UKJ_1145/19	Phe397Leu	Ala230Thr, Asp441Gly	No amplificates	[17,23]
		VIII	_UKJ_262/21		Ala448Thr	Tyr444His	No amplificates	[17,23]
		VIII	_UKJ_476/21		Ala448Thr	Wild-type	Type II	[17,23]
	*mentagrophytes*	VII	_UKJ_1722/22		n.d.	n.d.	n.d.	Routine
		XXV	_UKJ_173/19	218292/17	n.d.	n.d.	n.d.	[13]
	*benhamiae*	*v. benhamiae*	_DSM_6916	_RV_26680	n.d.	n.d.	n.d.	DSMZ
		*v. luteum*	_UKJ_117/23		n.d.	n.d.	n.d.	Routine
			_UKJ_488/23		n.d.	n.d.	n.d.	Routine
	*erinacei*		_UKJ_314/23		n.d.	n.d.	n.d.	RV491/23
	*quinckeanum*	I *	_IHEM_13697		Wild-type	Wild-type	n.d.	BCCM
		II *	_UKJ_1506/20		Wild-type	Wild-type	n.d.	[29]
*Microsporum*	*audouinii*		_UKJ_317/23		n.d.	n.d.	n.d.	RV491/23
	*canis*		_UKJ_299/23		n.d.	n.d.	n.d.	Routine
			_UKJ_300/23		n.d.	n.d.	n.d.	Routine
*Epidermophyton*	*floccosum*		_UKJ_317/23		n.d.	n.d.	n.d.	RV491/23

* *T. quinckeanum* isolates differ in ITS region as described [29]. ITS subtype of the *T. mentagrophytes* complex based on the mentioned nomenclature [13]. IHEM strains were derived from Belgian Coordinated Collections of Microorganisms (BCCM), CBS strains were stored at Westerdijk Fungal Biodiversity Institute, and DSM strains were obtained from the DSMZ-German Collection of Microorganisms. DNA sequences that were not determined were abbreviated as n.d.

## Data Availability

DNA sequence data were stored in GenBank with Acc. No. PV768551-PV768562 and GenBank Acc. No. PV779182-PV779185.

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
