# Peer review of "Significant Impact of Growth Medium on Itraconazole Susceptibility in Azole-Resistant Versus Wild-Type Trichophyton indotineae, rubrum, and quinckeanum Isolates"

_ijms, 2025, doi:10.3390/ijms26157090_

Round 1

Reviewer 1 Report

Comments and Suggestions for Authors

The paper could be of interest; however, as it is currently presented, it creates confusion for the reader and should be modified. The title clearly refers to the challenges in the cultivation of T. indotineae in relation to the assessment of azole susceptibility, but the text extensively discusses the genetic composition of the fungus. I assume the authors are suggesting that the presence of certain genes is responsible for its resistance. However, are these genes actually causing the cultivation issues?

The authors should focus on cultivation methods, the specific challenges encountered, any novel cultivation techniques, and demonstrate that these new methods allow for a more accurate in vitro assessment of drug susceptibility.

The presence of specific resistance-related genes could be the subject of a separate paper.

Moreover, the introduction is overly long and should be streamlined to better reflect the objective of the study. The authors dwell too much on Aspergillus genes.

The Introduction should clearly define the issue encountered with the cultivation of the dermatophyte and explicitly state the aim of the study (i.e., what the authors intend to demonstrate).

In the Materials and Methods section, the procedures should be described in detail. If the hypothesis is that the genetic composition influences the behavior of the fungus, then the experimental procedures related to the genomic analysis should be reported at the end of this section.

The same structure should be followed in the Results: first, present in detail the findings related to cultivation (e.g., temperature, media, etc.), followed by the genomic results.

The Discussion should focus on comparing the authors’ findings with those previously reported in the literature.

Reviewer 2 Report

Comments and Suggestions for Authors

This referee suggests that the authors include in the abstract information about the pathologies produced by these dermatophytes and reduce the too detailed information that will be included in the introduction. In addition, the abstract introduces many variables (media, temperature, genotype) without clearly delineating the primary hypothesis or conclusions. 

Further observations are indicated by points

-The study uses ICâ‚…â‚€ values, whereas EUCAST and CLSI protocols focus on MIC values for clinical breakpoint assessments.

-No justification is provided for choosing ICâ‚…â‚€ over MIC.

-The paper frequently restates that SG improves separation, but does not investigate why. Is itraconazole more soluble? Does pH drift lower in SG medium?

-Numerous grammatical issues, overly long sentences, and inconsistent formatting

-Inconsistent italicization of genus/species names.

For the reasons stated above, the manuscript requires detailed revision to make it clearer and give it greater value.

Reviewer 3 Report

Comments and Suggestions for Authors

This study evaluated the effects of medium composition, incubation temperature, and spore concentration on itraconazole susceptibility testing in 38 dermatophyte isolates using microplate laser nephelometry (MLN). Sabouraud glucose medium combined with lower spore concentrations provided better differentiation between resistant and sensitive strains, revealing significant influences of temperature and media on ICâ‚…â‚€ outcomes. Genotypic analysis identified resistance-associated mutations, emphasizing the need for species-specific ECOFF values to improve susceptibility testing protocols. This study is comprehensive, well-organized, and suitable for publication in IJMS after revision. Herein, I provide some comments for the revision.

  1. In the introduction, some paragraphs contain only 2 or 3 sentences. Please combine them with other paragraphs.
  2. Similarly, in the discussion, please combine paragraphs containing only 2 or 3 sentences with other paragraphs.

Round 2

Reviewer 2 Report

Comments and Suggestions for Authors The authors have taken note of the observations made and corrected the manuscript,
making it clearer and easier to read. It can be published in this form.
